cultural adaptation; post-traumatic stress disorder; Botswana; intervention: psychological; psychoeducation; breathing retraining

**Corresponding author:**
Keneilwe Molebatsi;
Emails: keneilwekmolebatsi@gmail.com; molebatsik@ub.ac.bw

# Cultural adaptation of a brief psychological intervention for PTSD in severe mental illness: a Botswana context

Keneilwe Molebatsi[1,2] , Tsholofelo Lobakeng[3], Lauren C. Ng[4] and Bonginkosi Chiliza[1]

[1]Department of Psychiatry, Nelson R. Mandela School of Clinical Medicine, University of KwaZulu-Natal, Durban, South Africa; [2]Department of Psychiatry, Faculty of Medicine, University of Botswana, Gaborone, Botswana; [3]Department of Psychology, Sbrana Psychiatric Hospital, Lobatse. Botswana and [4]Department of Psychology, University of California Los Angeles, Los Angeles, CA, USA

## Abstract

High rates of trauma exposure among patients with severe mental illness (SMI) in Botswana highlight the need for appropriate interventions. Culturally adapted interventions have been reported to be more acceptable, effective and feasible. This study aimed to culturally adapt the Brief Relaxation, Education and Trauma Healing (BREATHE), a brief psychological intervention to treat post-traumatic stress disorder (PTSD) among people with SMI in Botswana. The cultural adaptation process followed the steps outlined by previous research. They included a community assessment to identify needs, selecting an appropriate intervention and consultations with experts and stakeholders. Individual interviews and focus groups were conducted with patients living with SMI and mental health professionals, respectively, to inform domains of the intervention to be adapted. BREATHE was adapted to be culturally congruent to Botswana by following the ecological validity model framework and using data from the interviews. Examples of the adaptation include language that was translated to Setswana, and spoken English and the content that was revised to reflect the traumatic experiences and demographics of the Botswana population. The study underscores the utility of using evidence-based frameworks to culturally adapt interventions. The adaptation process resulted in a culturally relevant BREATHE for patients with comorbid PTSD and SMI in Botswana.

## Impact statement

Patients with severe mental illness (SMI) are at an increased risk of exposure to traumatic experience and, consequently, the development of post-traumatic stress disorder (PTSD). That notwithstanding, there is limited research in Africa on PTSD interventions suited for patients with SMI in this setting. Previous research findings have consistently demonstrated that interventions are more acceptable and feasible when they are culturally appropriate. In this article, we report on the steps followed to culturally adapt a brief psychological intervention for people with comorbid PTSD and SMI called Brief Relaxation, Education and Trauma Healing (BREATHE) for use in Botswana. The authors have detailed an evidence-based stepwise approach to culturally adapt the BREATHE intervention. The steps followed included community assessment, understanding the intervention, consulting experts and stakeholders, deciding what needs to be adapted and adapting the intervention. The community assessment unearthed mismatches between the original BREATHE intervention and the Botswana culture, thus highlighting and supporting cultural adaptations of interventions before implementation in settings different from where they were developed. The study utilized previously used and tested frameworks to inform domains of the intervention to be adapted, such as language, persons, metaphors, content and contexts, to make it culturally appropriate while preserving core components of the intervention, namely psychoeducation and breathing retraining. The study will play an integral role in informing methodologies for enhancing cultural congruence of psychological interventions without losing the core components, thus maintaining intervention efficacy.

## Introduction

### Trauma and post-traumatic stress disorder among individuals with severe mental illness

Severe mental illness (SMI) refers to chronic psychiatric disorders that significantly impair functioning (Regev and Josman, 2020). These include schizophrenia, schizoaffective disorder,

major depressive disorder and bipolar disorders (WHO, 2017). Post-traumatic stress disorder (PTSD) is highly prevalent among patients with SMI compared to the general population, with previous research reporting rates of PTSD in this group at 30% (Mauritz et al., 2013). This high prevalence is also observed in specific SMI disorders. For example, Zammit et al. reported PTSD rates of 31% and 39% among patients with psychotic and affective disorders, respectively (Zammit et al., 2018). Similarly, a recent review reported a PTSD prevalence of 13% in patients with psychotic experiences (Seong et al., 2023), while Nabavi et al. (2015) reported a pooled prevalence of 10.8% in bipolar disorder (Nabavi et al., 2015). In addition, another review reported PTSD prevalence as high as 57% in schizophrenia (Seow et al., 2016). These rates are notably higher than the estimated 4% prevalence in the general population (Koenen et al., 2017) and underscore the need for evidence-based interventions (EBIs) for treatment of PTSD among patients with SMI.

There are no epidemiological studies on PTSD in Botswana, despite widespread trauma exposure, including sexual violence (Becker et al., 2019; Bojosi et al., 2024; Ramabu, 2020), intimate partner violence (Barchi et al., 2018) and wildlife conflict (Buchholtz et al., 2023). Furthermore, Botswana is facing a rise in gender-based violence (Mooketsane et al. 2023), with studies reporting sexual abuse among 17% of youth living with human immunodeficiency virus (Molebatsi et al., 2024) and 37% of females with SMI (Bojosi et al., 2024).

Anecdotal evidence suggests that a significant proportion of patients admitted with SMI at the only tertiary psychiatric hospital in Botswana have experienced trauma and/or post-traumatic stress symptoms, consistent with global evidence, indicating that rates of trauma and PTSD are higher in this population than in the general population. These findings are supported by a recently published study, which reported that 93.8% of patients with schizophrenia at the psychiatric hospital had experienced at least one adverse childhood experience (ACE), with 56.3% having experienced four or more ACEs (Bojosi et al., 2024), further highlighting the need for targeted trauma interventions for individuals with SMI.

### Interventions for PTSD comorbid with SMI

Several treatments for comorbid SMI and PTSD have been studied (Grubaugh et al., 2021). One such intervention is the Brief Relaxation, Education and Trauma Healing (BREATHE), an evidence-based three-session intervention for PTSD in patients with SMI, which was developed in the United States (Nishith et al., 2015).

The core components of BREATHE are psychoeducation and breathing retraining. Patients are educated about trauma and PTSD, and breathing retraining skills are imparted for self-management of anxiety. The BREATHE intervention manual, handouts and video are utilized in the delivery of the intervention (Mueser et al., 2015). The video used in psychoeducation illustrates different potentially traumatic events, post-traumatic symptoms and complications. For example, the first session starts with the therapist introducing themselves, explaining why the patient is undergoing BREATHE and what the intervention entails. The therapist then identifies the traumatic event causing the patient the most distress. The patient then watches a selected section of the video, after which questions provided in the manual are used to guide discussions between the therapist and the patient. Questions include, "*When X happened, (referring back to a trauma that they described) do you remember having feelings of overwhelming fear, helplessness or horror?*" and "*Have you ever experienced reactions like those*

described on the video? (e.g., depression, fear, panic attacks)." To end the session, the patient is taught breathing retraining and is given a handout to practice at home. The next session starts with a homework review and proceeds to the video and discussions as outlined in the manual.

BREATHE was developed as part of a 4-year randomized control trial comparing the 12- to 16-week cognitive behavioral therapy (CBT) for PTSD program with BREATHE. Although the CBT group had better PTSD outcomes than BREATHE, there were no differences between CBT and BREATHE on depression, post-traumatic cognitions, functioning and quality of life (Mueser et al., 2015). The findings from the study showed higher rates of retention with BREATHE (95%) than with CBT (75%); notably, retention has been found to be indicative of therapeutic success (Nishith et al., 2015).

BREATHE has been found to be more cost-effective than the trauma-focused CBT program that was developed for patients with SMI (Slade et al., 2017), thus making it suitable for low- and middle-income countries such as Botswana.

While BREATHE has demonstrated effectiveness in other settings, its cultural appropriateness for Botswana requires careful consideration. As with any EBI, cultural adaptation is key to ensuring that it is effective and well-received in a new context.

### Cultural adaptation of EBIs

There are no studies in Botswana on the efficacy, acceptability and feasibility of evidence-based trauma interventions. This represents a critical research gap in understanding how well these interventions work in the context of Botswana's healthcare system, cultural norms and available resources.

While some scholars contend that using interventions tested elsewhere is more efficient than developing new ones (Moore et al., 2021), research has demonstrated that intervention effectiveness, acceptability and feasibility depend on context (Moore et al., 2021; Nagayama et al., 2016; Skivington et al., 2021). Therefore, for an intervention like BREATHE to be successful in Botswana, it must be adapted to align with the cultural, social and healthcare realities of the country.

Program adaptation was originally defined in 1962 as "*the degree to which an innovation is modified in the process of its adoption and implementation*" (Rogers, 1962) and could involve deleting, adding or just modifying some of its original components (Chen et al., 2013). Cultural adaptation ensures cultural congruence when interventions are used in populations or settings different from the original population or setting where the EBI was developed. Bernal et al. (2009, p. 362) define cultural adaptation as the "*systematic modification of an evidence-based treatment to account for language, culture, and context in a way that is consistent with the client's cultural patterns, meanings and values*" (Bernal et al., 2009). Culturally adapted interventions are more acceptable, effective and feasible (Escoffery et al., 2018; Hall et al., 2016) due to the integration of local idioms, spiritual beliefs, cultural norms and population needs during the cultural adaptation process (Li et al., 2017).

Escoffery et al. (2019) identified 11 key steps in the adaptation of public health EBIs. These include conducting a needs assessment, exploring the intervention's theory, selecting an intervention based on the community needs, consulting experts and stakeholders, adapting the intervention, training personnel, testing the intervention and implementing and evaluating the adapted intervention (Escoffery et al., 2019).

Two frameworks were used to inform the design of the cultural adaptation of the BREATHE intervention.

The ecological validity model (EVM) (Bernal et al., 1995) proposes eight dimensions to be considered during adaptation. These include language, persons, metaphors, content, concepts, goals, methods and context. Language is crucial in conveying emotional distress across cultures. Bernal et al. argue that interventions not offered in the native language may not be received as intended even if patients understand the language due to potential comprehension deficits arising from cultural nuances. The persons dimension considers cultural differences between patients and therapists, while metaphors involve incorporating local expressions or idioms in interventions. Content and context involve incorporating the political, social and economic factors of a population into an intervention. The EVM suggests that goals should align with both the patient and therapist's perspectives, and methods to achieve them should be culturally sensitive.

The cultural sensitivity model identifies two culturally sensitive dimensions: surface and deep structure. Surface structure involves incorporating external cultural characteristics, such as language, food, location and clothing, into interventions, while deep structure addresses social, historical and psychological factors that influence health behavior.

This article reports the methodology employed to culturally adapt the BREATHE intervention to treat PTSD among people with SMI in Botswana. The current study followed the key adaptation steps outlined by Escoffery et al. (2019) and drew from EVM and the surface structure of the cultural sensitivity model.

## Methods

### Study setting

Our study protocol providing the theoretical and methodological rationale of the proposed study has been previously published (Molebatsi et al., 2021). Data were collected at Sbrana Psychiatric Hospital, Botswana's only referral psychiatric hospital, which serves patients from diverse cultural backgrounds across the country.

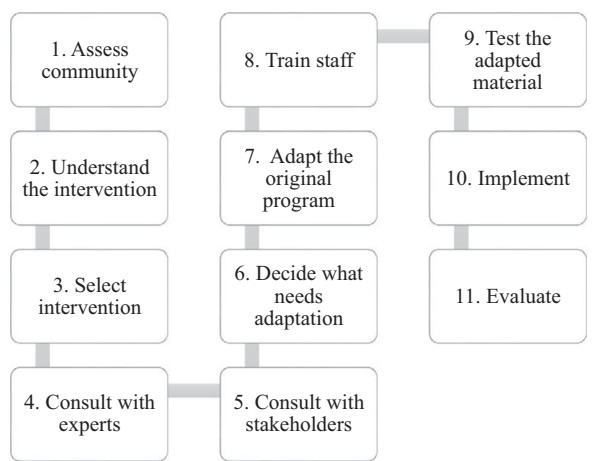

**Figure 1.** Key adaptation steps outlined by Escoffery et al. (2019).

### Steps followed

The study procedures followed Steps 1–7 as outlined by Escoffery et al. (2019) in Figure 1.

### Step 1: Community assessment

To understand the population before adaptation (Moore et al., 2021), we conducted in-depth interviews with 20 purposefully recruited patients with SMI. Patients were assessed for capacity to consent using the University of California, San Diego Brief Assessment of Capacity to Consent (UBACC) (Jeste et al., 2007). To determine trauma exposure, patients were asked to describe any traumatic events they had experienced. Sample questions from the interview guide included:

*"Can you tell me what the word 'trauma' or 'stressful event' mean to you?"*

*Have you ever considered an event or experience that you or another person has experienced (The event could have happened to you or a close family member or friend, or you could have witnessed it happening to someone else) to be traumatic?* (If yes, ask for examples).

In addition, five focus groups were conducted with diverse mental health professionals, including social workers, clinical psychologists, psychiatric nurses and psychiatrists. The number of groups was guided by Guest et al. (2017) findings that 90% of all themes are discoverable within three to six focus groups (Guest et al., 2017). Prior research recommends six to eight participants per focus group (Gill et al., 2008); while we aimed for six, the average was four per group, with numbers ranging from three to five. Some participants withdrew at the last minute due to scheduling conflicts arising from the hospital's demands. The researchers used purposive convenience sampling to recruit mental health care professionals, focusing on those directly involved in patient care and available at the time of recruitment. The focus group aimed to assess clinicians' views on the role of trauma in managing patients with SMI, explore traumatic experiences among their patients and identify key features of interventions deemed feasible and acceptable in our setting. The interview guide was adapted from a similar study that the third author conducted in Ethiopia (Ng et al., 2021). Examples of questions asked in the focus groups were:

*Sometimes patients experience traumatic events. One of the things we are interested in is how to help people cope with trauma in their lives. What are your thoughts about that?*

*(Probe- is trauma an important issue in your setting?)*

Interviews were conducted in English and transcribed verbatim. The lead author verified transcription accuracy by comparing audio recordings with the text. The first and second authors then performed thematic analysis (Braun and Clarke, 2006).

### Steps 2 and 3: Understanding and selecting the intervention

A literature search was conducted to explore PTSD interventions for patients with SMI, and experts, including the third author, were consulted.

### Steps 4 and 5: Stakeholder/expert engagement

The focus groups in Step 1 also explored potential barriers and facilitators to the intervention, including factors that could impact its feasibility, acceptability and effectiveness, as well as those that could promote its adoption in our setting. This step also sought recommendations on how to optimize the intervention for our

setting. For example, questions addressed the best location for delivering the intervention and which professionals the experts believed would be most suited to administer it.

### Step 6: Deciding what needs to be adapted

Following the community assessment and stakeholder engagement, the researchers used the EVM model to adapt the content of the BREATHE intervention, focusing on language, persons, metaphors, content and context.

### Step 7: Adapting the intervention

The BREATHE adaptation was guided by the EVM dimensions and incorporated the external surface structure as per the cultural sensitivity model. Findings from individual interviews and focus groups informed content adaptation, including what to adapt and how to do it.

A technical working group was formed, consisting of a clinical psychologist, a patient with comorbid PTSD and SMI, a psychiatric nurse and a social worker, none of whom had been interviewed previously to provide oversight and ensure fidelity throughout the adaptation process. (Rolleri et al., 2014; Solomon et al., 2006; Wingood and DiClemente, 2008).

## Results

The findings of this study map onto the steps outlined in Section "Methods."

### Step 1: Community assessment

#### Individual interviews

A total of 26 patients with SMI were approached during recruitment. One patient lacked the capacity to consent, as assessed with the UBACC (Jeste et al., 2007). Five declined participation due to the interview length. Ultimately, 20 participants consented, all reporting exposure to a traumatic event, thus meeting the inclusion criteria.

Participants originated from 9 out of the 10 administrative districts in Botswana (Statistics Botswana, 2022), representing the country's diverse cultural backgrounds. The interviewed patients, age range 18–65 years, were 55% male ($n = 11$). The most common diagnosis was schizophrenia (60%), and the least was schizoaffective disorder (5%). Participants reported traumatic events consistent with DSM-5 Criterion A for PTSD, including the death of a loved one, sexual violence, physical violence and natural disasters. They also reported events outside this criterion, such as mental health service-related stress, conflictual relationships, financial stress, toxic work environments and spiritual or cultural stressors within the context of indigenous or spiritual beliefs, such as struggles emanating from refusing to accept an ancestral calling.

#### Focus group

We conducted five focus groups with a psychiatrist, three clinical psychologists, five psychiatry residents, one medical officer, four nurses, two social workers, two counselors and two intern counselors. Participants' ages ranged from 22 to 56 years, and there was equal representation of gender.

While there was agreement that trauma is important in managing patients with SMI, two divergent views emerged on screening and treatment. Some clinicians believed trauma is adequately screened

and treated, while others felt there are no standard screening measures, leading to potential underdiagnosis of PTSD.

"*Every time when you see patients, some patients would have experienced some traumatic events and you need to attend to them and attend to their experiences.*" (MH01F5)

"*Uh, I'd actually like to disagree. I don't think that trauma is a consideration within this specific facility. As a professional individually, I think it is an important thing to consider…*" (MH03F5)

Moreover, there was a consensus among the clinicians that there are inconsistencies in screening for PTSD. They attributed this to the lack of established standard screening measures as well as clinicians' perceptions of what constitutes a significant traumatic event.

"*We don't have that [standardized PTSD screening]… for PTSD we don't.*" (MH04F2)

"*So, patients would complain of certain things that you wouldn't believe are very traumatic. But the problem with us is we don't go ahead to assess for the possible outcome of the trauma like PTSD because it doesn't fit the criteria for them to have PTSD from that traumatic event…*" (MH04F2)

Similar to patients, experts recalled traumatic experiences matching DSM-5 PTSD Criterion A, as well as other events such as animal attacks, inheritance disputes, combat injuries, forced medical treatments and traditional religious beliefs, which were incorporated in adapting the content of the intervention.

### Steps 2 and 3: Understanding and selecting the intervention

Interventions identified in the literature review as having been evaluated for reducing PTSD symptoms in patients with SMI were prolonged exposure, eye movement desensitization and reprocessing and CBT (Grubaugh et al., 2016; Mueser et al., 2015; Van Den Berg et al., 2015; Van Den Berg and Van Der Gaag, 2012). Interventions ranged in duration from three to 15 sessions. There are limited trained mental health experts in Botswana (Opondo et al., 2020); thus, the researchers deemed a brief intervention appropriate to mitigate against the scarcity. The BREATHE intervention is a three-session psychological intervention that was developed to treat PTSD among people with SMI in Botswana. BREATHE was chosen for its feasibility in resource-limited settings and its ability to be delivered by trained lay workers, thereby addressing the challenges posed by the limited availability of specialized mental health care.

### Steps 4 and 5: Stakeholder/expert engagement

Experts and stakeholders added to the list of traumatic events commonly treated in Botswana, as detailed in the results under Step 1. In doing so, they facilitated the next steps by discussing the context and content of what needed to be adapted.

Regarding the intervention, the participants expressed a range of attitudes. On the one hand, they expressed a sense of appreciation for the BREATHE intervention, praising its brevity and emphasis on psychoeducation and breath work. They also appreciated the researchers' efforts to culturally adapt the intervention and even cited the benefits of cultural adaptation.

"*What I liked about it is how they include the breath work to target the hyper arousal that you see in patients that have trauma.*" (MH03F5)

"*I like the fact that it's a combination of management and psychotherapy… the fact that the research is part of it is about the cultural*

*adaptation and the psycho education part of it that you know, recovery it aids for a patient to understand fully what they're going through for them to recover… The cultural adaptation, I like it because of some of these patients the way they present if you are not culturally oriented you'll miss certain things because, an old lady who will come and tell you that my heart skips faster this happened and literally, they associate their illness with the with the heart literally, but those are things that come, I think, for this for this intervention on the psychoeducation aspect where you will be explaining whether the explanations of what they went through and how it even leads to them experiencing the palpitations that make them think this is heart had related. I like that holisticness about it, but most basically the psychoeducation part and the cultural adaptation part of it.*" (MH04F5)

On the other hand, they expressed skepticism owing to potential barriers such as lack of resources, such as television sets in clinics, lack of skilled personnel and challenges with dual relationships.

"*[They] do not have the TVs and all those things, so that means the TVs will need to be provided in clinics and by who.*" (MH02F4)

"*I think some of them would be possible lack of training adequate training for dealing with those problems.*" (MH02F1)

### Steps 6 and 7: Deciding what needs to be adapted and adapting the intervention

Following the EVM dimensions, this step relied on data from the individual interviews, focus groups and guidance from the technical working group.

#### Context

An intervention adaptation must be based on an understanding of the context in which it will be applied. Focus groups and the technical group described Botswana as a mental health expert-limited environment. There were differing opinions on potential providers; majority were of the belief that unemployed psychology degree holders would be the most appropriate interventionists.

*I think it should also be the government's role and we can even vote that these are our students completing degrees in psychology, who failed to go further, they could be absorbed somehow, trained and be put in better positions to provide such services. (MH01F1)*

However, some believed that nurses would be the most appropriate interventionists given their roles as the primary point of contact in primary healthcare posts and their widespread presence across the country. "*I think nurses for sure, because they're available like throughout the entire country or like local facilities as well.*" (MH03F5)

On the contrary, some experts argued that an appropriate interventionist should have a background in counseling; hence, they deemed social workers, psychologists and psychiatrists as the most appropriate interventionists.

"*…Overall I think psychologists and psychiatrist, maybe I don't know to what extent, but maybe social workers I think.*" (MH02F1)

There seemed to be a consensus about primary health being the best place for the intervention. Thus, the intervention would be best suited to be administered by first-contact providers at the primary mental health level.

"*Primary care remains the backbone of any health system and if you were to look at the relationships between the patients and the primary care giver and patients and tertiary care giver you would find that the better relationship is with the primary care giver because that where*

*they spend a lot of time that's where they establish long time rapport…*" (MH05F1)

The inconsistencies in clinicians' perceptions regarding the best-suited provider for the intervention underscore the importance of evaluating adapted interventions. While the decision on the most appropriate provider for delivering the intervention in Botswana will be made after testing and evaluating the adapted intervention, psychology graduates were proposed by the majority, while nursing staff were preferred because of their role as patients' first contact in health care.

#### Language

Focus group members and technical group members both believed that it was imperative to translate the intervention into everyday spoken Setswana and English without technical jargon.

"*And the medium of communication, because if they do then, words like somatic…it's not easy to explain them in Setswana*" (MH01F3)

The intervention manual was translated into Setswana and Botswana English and verified for thematic equivalence by the technical working group. The group reviewed the adapted manual and script against the originals in a focus group led by the first author, with disagreements resolved by consensus.

In the video, the psychologists' comments were clear enough for laypeople, so they were translated into Setswana without alteration. Participants' stories (see Section "Content") were modified to reflect local traumatic experiences and sequelae. The translator's work was revised to sound more natural, as the technical team felt it lacked everyday language familiarity. For example,

Original: *The pain is deep; I mean it's like its winds around your bones and goes all the way and grabs your toes and holds on…*

Adapted: *The pain was just too much, filled all my life. It overcame me, taking out any little joy I had…*

The video was recorded after the first technical group meeting, reviewed by the working group and feedback was analyzed to inform further revisions. Final adaptation decisions were made during the meeting.

#### Content

The original video shows clients and a psychologist. Clients tell their stories of lived trauma experiences, and the psychologist conceptualizes or explains them from a psychological perspective. As the narratives are specific to the American context, the technical group and the experts believed they mostly would not resonate with a typical Motswana (an individual from Botswana) due to cultural variance.

"*But we have to understand the culture in which we operate and some of you know, live the cultural formulation really. So, for a lot of people to be bewitched is a real thing. It's a belief that is, you know, acceptable within their communities, and there are certain people that are known for witchcraft and to be bewitched is not bizarre. So, when he, but I think it was a she who came with the evidence of being bewitched, but I am not finding any help, but as for her it's not the issue of going for revenge or whatever but then she is then presenting with symptoms of trauma right, and distress from the belief that they are bewitched and this is not psychotic in nature.*" (MH02F5)

The content was adapted based on the findings. Examples are depicted in Table 1.

#### People

In the original video, white Americans are depicted. To increase the acceptability of the intervention, it was recommended that the

**Table 1.** Illustration of content adaptation

| Narratives from the original video |
| --- |
| • I experienced a lot of depression I first noticed it when I was 13 and had a prolonged depression for 6 months and more and continued to have increasingly severe depression throughout my teens<br>• I have startling response, like oh my God, if somebody bangs the door and I don't know what's coming, I fly off the chair, I have one of the worst startle response in the whole world |
| **Narratives from the adapted video** |
| • The clash between my ancestral calling, my family's Christian beliefs, and them forcing hospital medicine created a war within me, leading to a mix of bad feelings in me…. It got so bad that I ended up in a mental hospital, and it all started with me being put in handcuffs, which was really scary.<br>• Growing up I really believed my mother hated me. I was the only boy and I remember as early as 8 years my mother would tie me to a tree when I had done something that she didn't like and beat me up, I still have some scars from those days. |

video depict Batswana (people from Botswana) with typical physical appearances. The majority of Botswana's population is Black, so we adapted the video to show Black people from various Botswana tribes with varying accents and ages ranging from 25 to 55 years old.

> "And I think in addition to that language aspect, just things that are familiar and not too foreign like I learn from seeing things that are familiar. So, like even this education, whoever will be, the video that you are watching locally, shot in phenotype that is very unfamiliar to me or uncomfortable for me. I was gonna feel like, it's too Western is a concept from wherever. Like just normal people explaining normal things in a language that is normal to me and look like me." (MH02F5)

### Concepts

According to the participants, Batswana typically describe physical symptoms rather than their emotional reactions, suggesting that concepts that reflect physiological arousal could be effective in describing their experiences.

> "An old lady who will come and tell you that my heart skips faster this happened and literally, they associate their illness with the with the heart literally." (MH04F5)

These concepts were incorporated when adapting the video, for example:

Adapted: *These made me tense, heart beating fast, teeth clenched and fists tight…feeling hot and cold at the same time. It would be like my hair is shrinking…*

### Methods

In the experts' opinion, it might be very difficult to find television sets, and some people would not enjoy watching videos. A few suggested a non-digital dissemination method for the intervention.

> "For me, the thing about mode of delivery, I don't know whether people will watch TV and then feel that the things that they see on TV will help them, or maybe our attitude to watch TV like I was watching soap and this is what happened, so am saying it from that point, because in primary health care you may be dealing with very traditional people." (MH04F1)

### Discussion

This study aimed to culturally adapt BREATHE, a brief psychological intervention for PTSD among patients diagnosed with SMI in Botswana. The adaptation process was guided by key adaptation steps outlined by Escoffery et al. (2019), while incorporating and using the EVM dimensions and surface structure of the cultural sensitivity model to inform intervention elements that should be adapted. The use of multiple frameworks enhanced the rigor and comprehensiveness of the cultural adaptation. The adaptation process was further enriched with data collected from potential end users of the adapted intervention and mental health professionals.

Some scholars posit that cultural adaptation can risk fidelity loss and undermine intervention effectiveness (Castro et al., 2010; Segrott et al., 2014). Their argument is rooted in the view that changes to the key components of the intervention can undermine the theoretical tenets of the intervention. Therefore, in this study, we sought to balance retaining the core elements of BREATHE (psychoeducation and breathing retraining) with the incorporation of essential cultural components into the adapted intervention.

To understand the unique needs of patients with comorbid SMI and PTSD in Botswana and ascertain whether the intervention was a necessity in our setting (Perera et al., 2020), the researchers interviewed SMI patients with lived experiences of trauma and mental health clinicians. Furthermore, the experts and the patients provided valuable information regarding traumatic experiences and post-traumatic sequelae in our population to guide adjustment of the characters and the storyline in the intervention, thus making the content, concepts and context of the intervention relevant to the setting.

The interviews were also beneficial in understanding community needs, which has been shown to be essential in the successful cultural adaptation of interventions (Gearing et al., 2013). Previous studies have highlighted the importance of community assessment in cultural adaptation, suggesting that failure to do so may lead to interventions that are less effective, less engaging and less sustainable for the target population (Burlew et al., 2020; Valdez et al., 2018).

In addition, previous literature has highlighted that the adaptation process of an intervention should carefully examine the target population's specific characteristics, such as cultural, socioeconomic and demographic factors, to ensure that the intervention is relevant and effective (Heim and Kohrt, 2019). In line with this, mental health clinicians pointed out the colloquial and everyday language, local perceptions of trauma and physical traits of the characters in the videos as critical elements to be considered. These findings align with previous studies that suggest a positive correlation between the extent of cultural adaptation and intervention efficacy (Shehadeh et al., 2016).

In our study, the experts suggested that primary health facilities were best suited for the intervention delivery aligning with previous research that reported a high prevalence of mental disorders in primary health care (Ansseau et al., 2004; Motsohi et al., 2015). Notably, the majority of patients with SMI view primary care as the cornerstone of their mental health care (Lester et al., 2005). This underscores the need to develop interventions suitable for administration at the primary care level in low- and middle-income countries, such as Botswana. The availability of primary health care facilities across the country supports the feasibility of scaling up the intervention. In addition, the provision of interventions at the primary care level aligns with the Government of Botswana's agenda for integrating mental health care into primary healthcare services (Ministry of Health and Wellness, 2020).

Both patients and experts expressed concerns about a video-based intervention due to the lack of resources, such as televisions in health facilities. This was an important consideration during the

adaptation as it could affect the reach of the intervention. To address this, the authors will explore low-cost alternatives, such as using mobile phones or tablet devices. Alternatively, the researchers may utilize non-video-based versions of BREATHE, which will also undergo cultural adaptation.

In line with global trends and World Health Organization recommendations to widen access to health care through task shifting (Okoroafor and Christmals, 2023; WHO, 2007), majority of the mental health clinicians proposed that Bachelor of Psychology graduates who are not yet clinical psychologists could be engaged in administering BREATHE. Given the limited postgraduate training opportunities in Botswana, many psychology degree holders remain unemployed, making this a viable solution to reduce the burden on the country's limited number of clinical psychologists. Moreover, the government of Botswana has programs such as the Botswana National Service Program, Tirelo Sechaba, which engages unemployed graduates to provide essential services to the community (Government of Botswana, 2022). This initiative could be leveraged in the implementation of BREATHE. This further underscores the suitability of BREATHE for Botswana.

A culturally adapted intervention must be tested (Perera et al., 2020) to evaluate the adequacy of the adaptation. The BREATHE intervention will be pilot tested among Batswana patients with comorbid PTSD and SMI to assess feasibility, acceptability and efficacy (Molebatsi et al., 2021); the findings from this pilot study will be used to inform any further modifications to the intervention.

## Strengths and limitations

The use of different previously well-researched cultural adaptation frameworks and steps in this study contributed to the rigor of the cultural adaptation process while preserving the core components of the intervention (Barrera and Castro, 2006; Bernal et al., 1995; Escoffery et al., 2019; Resnicow et al., 2000). In addition, participants recruited from the psychiatric hospital are representative of the nation as they came from various regions of the country, offering a fair representation of the culturally diverse nation. The diversity strengthened key cultural considerations to be included in the adaptation process. Involving end users, target population and those who will deliver the intervention across all stages of the adaptation process optimizes its cultural fit, enhancing acceptability as it addresses perceived needs (Racine et al., 2022).

The study has several limitations to consider. First, due to the study design and the limited sample size limits, the researchers may have missed vital trauma-related experiences and perceptions to guide the cultural adaptation process. However, focus groups with mental health clinicians revealed other experiences that may have been experienced by patients not included in individual interviews, thus widening the representation of the findings. The study did not include caregivers of the intended end users whose insights could have enriched the findings.

## Recommendations

Future adaptation studies should develop study designs to incorporate research on the impact of cultural adaptations on the theoretical foundations of the intervention. In addition, including caregivers of patients with SMI in the adaptation process may also yield a broader understanding of patient needs and cultural factors to enhance the adaptation process. Furthermore, caregivers may provide insights on relevant interventions and practical considerations for the successful implementation of adapted interventions.

## Conclusion

This study utilized a combination of evidence-based guidelines and frameworks to culturally adapt BREATHE, ending up with a culturally relevant version of the intervention for patients with comorbid PTSD and SMI. Elements of the intervention such as language, content and metaphors were adapted while maintaining fidelity to the core elements. The study lays a foundation for future cultural adaptations of psychological interventions in our setting.

**Open peer review.** To view the open peer review materials for this article, please visit http://doi.org/10.1017/gmh.2025.37.

**Data availability statement.** The data that support the findings of this study are not available in order to maintain the confidentiality of the study participants.

**Author contribution.** Conception and design of the study: K.M., L.C.N., B.C. Data collection and analysis: K.M., T.L. Manuscript write-up and editing: K.M., T.L., L.C.N., B.C. Critical comments: L.C.N. Study supervision: L.C.N., B.C.

**Financial support.** The study was funded by the University of Botswana, training department.

**Competing interests.** The authors declare that they have no competing interests.

**Ethics statement.** The ethical approval of this study was granted by Ministry of Health, Botswana (HPDME 13/18/1), and the Institutional Review Boards of the University of Botswana (UBR/RES/IRB/BIO/179) and the University of Kwazulu Natal Biomedical Research Ethics Committee (BREC983/2020). The participants signed an informed consent document before participating and were notified that they were free to withdraw from the study at any time and it was entirely voluntary for them to participate. The audiotapes were password-protected and were safely stored by the first author. To maintain participant anonymity, the participants were pseudonymized, and identifying information was not included in the transcript.

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
