## [Reviewer Report]

This paper describes the adaptation of a brief intervention for posttraumatic stress disorder (PTSD) in people with the severe mental illness living in Botswana. The paper addresses an important topic given the high rates of PTSD in people with severe mental illness reported in many other countries, but the lack of treatment for the disorder in routine practice. The paper also addresses practical considerations of implementing and intervention for PTSD that is culturally adapted yet appropriate given the limited resources available in a low income country. The paper is relatively clear and straightforward, but could be improved by attention to points raised below.

1. At the beginning of the introduction, the authors referred to high rates of trauma in people living in Botswana, and point to the example of being attacked by wildlife. While rates of such trauma are undoubtedly higher in Botswana than many other countries, especially in high income countries, I would suspect that there are even higher rates of interpersonal trauma, as has been reported in most other countries, including both low and middle income and high income countries. Is anything known about the rates of interpersonal trauma in people living in Botswana?

2. In the second paragraph on page 3, the authors write “Anecdotal evidence suggests that a significant proportion of patients admitted with severe mental illness…” Admitted to where?

3. At the end of the sentence referred to in the previous point, the authors state that rates of trauma and PTSD symptoms are “comparable to global evidence.” This needs to be clarified. Do the authors mean that the rates of PTSD in people with severe mental illness are higher than in the general population, similar to research in other countries?

4. In the last paragraph on page 3, after noting that some people contend that using interventions developed elsewhere is more efficient than developing new interventions, in the next sentence they state “Additionally, research shows that intervention effectiveness, acceptability, and feasibility depend on context.” Since this sentence qualifies the previous sentence by noting that attention to context is necessary when using the intervention developed in one place in another place, rather than beginning the sentence with “Additionally” it might be preferable to write something like, “However, research also shows…”

5. On page 4, the authors write “Escoffery and colleagues conducted a scoping review identifying 13 adaption frameworks…” Two sentences later they write “Several frameworks have been developed to inform the design of cultural adaption methodologies.” Are these the same frameworks described in the scoping review previously mentioned?

6. Later, in that same paragraph on page 4, the authors write that “… interventions not offered in the native language may not be received as intended.” As written, this raises the question of how participants would understand an intervention if not delivered in their native language?

7. In the focus groups with mental health professionals, the authors describe discussing the issue of whether trauma was adequately assessed in people with severe mental illness in Botswana. Did these discussions also include the adequacy of the assessment of PTSD in such patients?

8. In the section describing the evidence for the BREATHE program the authors might also note that in the Mueser et al. (2015) study the BREATHE intervention was found to be more cost-effective than the full CBT program:

Slade, E. P., Gottlieb, J. D., Lu, W., Yanos, P. T., Rosenberg, S. D., Silverstein, S. M., Minsky, S., & Mueser, K. T. (2017). Cost-effectiveness of a PTSD intervention tailored for individuals with severe mental illness. Psychiatric Services, 68, 1225-1231.

9. On page 13, the authors describe the discussion in focus group participants about which health professionals would be most qualified to provide the BREATHE intervention. This discussion is a little confusing because early in the introduction the issue of task shifting was discussed as an important adaptation when implementing an intervention in a low resource country. This discussion in the introduction left the impression that task shifting might be employed when adapting and implementing the BREATHE intervention, but this appears to not be the case.

10. Related to the previous point, although different perspectives on who should deliver the intervention are described (above), the resolution to this issue in terms of modifying the intervention is not presented in the results, although it is mentioned in the discussion section. It would be preferable if the resolution to issues raised in the results section were also discussed in that section.

11. On pages 16 and 17, it is reported that the experts were concerned that it would be difficult to find television sets to play the videos on for the BREATHE, but the resolution to this issue was not discussed. It should be noted that similar or other concerns have been raised about the videos used in the BREATHE intervention (e.g., that the patients in the videos are not similar in age or race to patients where the program was being implemented) and that to accommodate this a non-video version of the BREATHE program also exists (Mueser & Gottlieb, 2025).

Mueser, K. T., & Gottlieb, J. D. (2025). Treatment of Posttraumatic Stress Disorder in Serious Mental Illness: The Cognitive Restructuring Program. American Psychological Association.

---

## [Reviewer Report]

This study aimed to culturally adapt the BREATHE intervention from the United States to Botswana. BREATHE is a brief psychological intervention to “treat” PTSD amongst people with severe mental illness. The cultural adaptation process followed three primary steps:

1. Community assessment to identify needs

2. Selecting appropriate intervention

3. Consultations with experts and stakeholders about the potential intervention

Individual interviews and FG discussions held with #1 and #3 above

OBJECTIVE - Full research articles: is there a clear objective that addresses a testable research question(s)?

Yes - there is a clear objective

DESIGN - Is the current approach appropriate for the objective?

Yes - the approach is appropriate

EXECUTION - Are the experiments and analyses performed with technical rigor to allow confidence in the results?

The experiments and analyses appear to have been performed appropriately, however it is difficult to follow several of the sections and steps to have confidence in the results.

STATISTICS - Is the use of statistics in the manuscript appropriate?

N/A

INTERPRETATION - Is the current interpretation/discussion of the results reasonable and not overstated?

The discussion reads more like a methods section in that it describes what was done. However, it does not move enough beyond restating what the researchers did and what the results were.

OVERALL MANUSCRIPT POTENTIAL - Is the current version of this work technically sound? If not, can revisions be made to make the work technically sound?

While the majority of the content is contained in the manuscript, the flow and sequencing are very confusing, which limit the strength of the study. Several sections are missing transitions, so that new concepts come out abruptly. To make the work technically sound, sections would need to be moved around and thoroughly rewritten for clarity and to better give context around why a concept is important.

Other notes:

• The manuscript will need thorough copy editing. There are many typos and grammatical errors throughout, which need to be corrected. For example, on page 1, line 13: PTSD should be spelled as posttraumatic stress disorder (all lowercase and posttraumatic is one word); on page 1, line 17, it should read “the authors have detailed an evidence-based”… on page 2, line 24, “they included a community assessment”… The manuscript also goes back and forth between current and past tense.

• Since the paper is about the adaptation of BREATHE, you should describe what BREATHE is in the beginning so that readers have context about it. Right now, you first describe it on page 10. In addition, since the study is about cultural adaptation, you will need to make it clear where BREATHE is being adapted from. Right now, you state that it is an “American context” on page 15.

• Readers will not necessarily know what Motswana and Batswana are (page 15). You will need to state what they are.

• Change “Caucasian Americans” to “White Americans.”

• Page 20, line 3: you cannot state that the patients at the hospital are representative of the country unless you have data to support this statement.

• You do not need to cite Bernal et al. and Resnicow et al. and Escoffery et al. every time you use them later in the paper after you have described them.

• What is EBI? (Page 4, line 10)?

• What is EVM? This acronym is not spelled out in the abstract and should be the first time you use it.

• I cannot see Figure 1 except for 11 pieces of text with small grey boxes in between.

• You should define severe mental illness when you first use it in the introduction (schizophrenia, bipolar disorder etc.) or at least give examples of SMI diagnoses. Right now, you define this on page 6.

• The introduction is confusing because you go from Botswana -> global -> Botswana. I would recommend starting with global statistics before you move to the context of Botswana.

---

## [Reviewer Report]

Thank you for giving me the opportunity to review this paper, which describes a cultural adaptation of a PTSD protocol in severe mental illness in the Botswana context, a unique context. It uses several models and details the process in a way that gives respect to the people and to the process, and makes one hopeful that it could help! I think this journal is the appropriate journal for the publication of such a paper. I do have several comments.

Regarding the structure of the article, some sections of the literature review are fragmented, and there are several short paragraphs. This hampers the fluency of the reading experience. For example, on page one, there are some treatments and some scholars presented, but with each having its own paragraph. On the other hand, some paragraphs are long. I think the rationale for the association between SMI and PTSD has to be strengthened, as well as the reasons for choosing the specific intervention.

The last stage of an adaptation is implementation and accepting the feedback of professionals and clients on the usability and feasibility of the intervention. This aspect is currently missing from the paper. Some preliminary feedback, from pilot clients and therapists, could validate and strengthen the paper.

In the discussion, in addition to talking about the strengths and limitations, I think there should be a better matching between the steps that were described in the results and the models that were presented in the introduction. Moreover, there should be a discussion and/or recommendations for future adaptations.

Maybe the authors could provide a link for one of the new movies they created, as an example?

---

## [Reviewer Report]

This paper is a revision of a paper describing the cultural adaptation of an intervention for the treatment of posttraumatic stress disorder (PTSD) in people with severe mental illness (SMI) living in Botswana. The authors have been responsive to the comments of the reviewers and thus the paper is much improved. The paper could be further improved by attention to the relatively minor points raised below.

1. The statement in the introduction that prevalence of PTSD in patients with psychotic experiences was 57.1% in the review by Seong et al. is not quite accurate. The Seong review indicated a lifetime prevalence of PTSD of 13% in patients with schizophrenia; the 57.1% prevalence refers to PTSD related to psychotic experiences (e.g., hallucinations, delusions) or treatment (e.g., involuntary hospitalization), which is not typically the focus of interventions for PTSD in the SMI population. More representative reviews of the prevalence of PTSD in SMI include:

Mauritz, M. W., Goossens, P. J. J., Draijer, N., & van Achterberg, T. (2013). Prevalence of interpersonal trauma exposure and trauma-related disorders in severe mental illness. European Journal of Psychotraumatology, 41, 19985. https://doi.org/10.3402/ejpt.v4i0.19985

Seow, L. S. E., Ong, C., Mahesh, M. V., Sagayadevan, V., Shafie, S., Chong, S. A., & Subramaniam, M. (2016). A systematic review on comorbid post-traumatic stress disorder in schizophrenia. Schizophrenia Research, 176, 441-451. https://doi.org/10.1016/j.schres.2016.05.004

Zammit, S., Lewis, C., Dawson, S., Colley, H., McCann, H., Piekarski, A., Rockcliff, H., & Bisson, J. (2018). Undetected post-traumatic stress disorder in secondary-care mental health services: Systematic review. British Journal of Psychiatry, 212, 11-18. https://doi.org/10.1192/bjp.2017.8

2. The statement on page 35 “Several treatments for comorbid SMI and PTSD have been studied” is followed by 4 references (e.g., Foa 2009; Markowitz et al., 2015), but none of these references are of studies of treating PTSD in people with SMI; all of the references refer to treatment of PTSD in the general population.

3. Page 35, line 28, the reference to Mueser et al. (2008) should be Mueser et al. (2015), the study in which the BREATHE intervention was first evaluated.

4. Page 36, line 36, change “…trauma interventions, this represents…” “…trauma interventions. This represents…”

5. Page 41, it would be helpful if the authors briefly elaborated on what they mean when they refer to “spiritual stressors” in the last line.

6. Page 42, line 24, when the authors refer to “potential underdiagnosis” presumably they are referring to potential underdiagnosis of PTSD, but this should be clarified.

7. Page 51, line 8, change “Notably, majority of…” “Notably, the majority of…”

Relatedly, in the same sentence on line 9, would it also be accurate to write that the majority of patients with SMI view primary care as the cornerstone of their mental health care?

8. Page 51, line 13, change “…at primary care…” “…at the primary care…”

9. Page 51, regarding the video component of the BREATHE program, it might be noted that versions of the program exist that have eliminated the video component. For example:

Mueser, K. T., Davis, K., Burke-Miller, J. K., Marcello, S., Gottlieb, J. D., Fraser, V., & Razzano, L. A. (Online 2024). Implementation of a brief treatment program for PTSD in persons with serious mental illness in a large mental health agency: The BREATHE program. Psychiatric Rehabilitation Journal. https://doi.org/10.1037/prj0000632

---

## [Reviewer Report]

The authors have been responsive to the comments made by the reviewer and no further revisions are needed.